

# *In situ* study of environmental factors (temperature and salinity) affecting cohort patterns and growth rates in *Ciona robusta*

Philjae Kim[1,2] and Seongjun Bae[3]

[1] DNA Analysis Division, National Forensic Service, Daegu, Republic of Korea
[2] Department of Marine Biotechnology, Kunsan National University, Gunsan, Republic of Korea
[3] Department of Ecology and Conservation, National Marine Biodiversity Institute of Korea, Seocheon, Republic of Korea

## ABSTRACT

Assessing and understanding the ecological impacts of marine invasive species is important for managing marine ecosystems, especially since their rapid growth, short reproductive cycles, and wide range of water temperature adaptability pose major challenges. In this study, conducted in Mokpo, South Korea, we explored the cohort dynamics and environmental influences on the ascidian *Ciona robusta*, which is considered a widespread invasive species. Through biweekly field surveys and quantitative measurements (dry weight, wet weight and body length) conducted from June to October 2022, we identified five distinct cohorts, challenging existing assumptions about lifespan and cohort patterns. All separation index value (which quantify differences between cohorts) exceeded 2, indicating clear separation of cohorts during the study period. The cohorts had a lifespan of between 4 and 10 weeks, much shorter than the previously reported maximum of 2 years. These differences suggest that local climatic conditions can have a significant impact on lifespan parameters. In addition, growth rates were significantly positively correlated with environmental conditions, particularly temperature. This highlights that while *C. robusta* growth rates are particularly sensitive to temperature changes, they showed relative tolerance to the salinity variations observed in this study. Therefore, this study contributes to the understanding of the population ecology of *C. robusta* in temperate marine ecosystems. In particular, it provides valuable insights for developing management strategies to mitigate the impacts of *C. robusta* due to climate change.

Corresponding author
Seongjun Bae, silverto@naver.com

## INTRODUCTION

To preserve marine ecosystems, understanding and evaluating the impacts of disturbance and pollution caused by marine invasive species is crucial (*Whitlatch & Bullard, 2007*; *Locke, 2009*; *Kanamori et al., 2017*). In addition, certain invasive species have negative impacts on human economic activities (*Schultz et al., 2011*; *Park et al., 2018*). Examples of fouling organisms, particularly those that live in a sessile life, include ascidians,

bryozoans, hydrozoans, barnacles, sponges, and mussels (*Bosch-Belmar et al., 2019*; *Shevalkar, Mishra & Meenambiga, 2020*; *Lins & Rocha, 2022*). Ascidiacea are a taxonomic group containing taxonomically and morphologically diverse species widely distributed in marine environments worldwide, with several species being common components of fouling communities (*Millar, 1971*; *Young, 1985*; *Sahade, Tatián & Esnal, 2004*). These invasive ascidians are one of the major causes of marine invasive species problems because of their rapid growth rate, short reproductive cycle, and lack of crucial predators (*Shenkar & Loya, 2008*; *Lynch et al., 2016*; *Kanamori et al., 2017*).

A primary life history characteristic of ascidians is that they are sessile as adults but, in contrast, are free-swimming as larvae (*Millar, 1971*). Larvae tend to settle near adult colonies (*Davis & Butler, 1989*). However, the dispersal pattern varies depending on the species, and in a study of *Didemnum vexillum*, larvae were found to disperse up to several hundred meters (*Fletcher, Forrest & Bell, 2013*). After development, larvae hatch from the eggs and disperse in search of suitable substrates for metamorphosis (*Chase, Dijkstra & Harris, 2016*; *Hirose & Sensui, 2021*). Following larval metamorphosis, they develop into adults through settlement (*Cloney, 1982*). The ascidian life cycle is influenced by substrate materials (*Anderson & Underwood, 1994*; *Chase, Dijkstra & Harris, 2016*), light conditions (*Nandakumar, 1995*), pH (*Jones, Holt & Chan, 2022*), temperature (*Kim et al., 2019a*; *Kim et al., 2019b*), and salinity (*Malfant et al., 2017*), which play a crucial role in the survival and growth of marine organisms.

*Ciona intestinalis* was reclassified into two distinct species based on molecular and morphological studies: *Ciona robusta* (formerly *C. intestinalis* type-A) and *C. intestinalis* type-B (*Brunetti et al., 2015*; *Gissi et al., 2017*). While *C. robusta* corresponds to the previously designated type-A, type-B remains classified under *C. intestinalis*, allowing for clear differentiation between the two types in various geographic regions (*Wilson, Murphy & Wyeth, 2022*). *Ciona robusta* is now recognized worldwide as a significant marine invasive species, prompting extensive research on its distribution and spread in various countries (*Bouchemousse, Lévêque & Viard, 2017*; *Shenkar, Shmuel & Huchon, 2018*; *Park et al., 2018*; *Kim et al., 2019a*; *Bae et al., 2023a*; *Bae et al., 2023b*). In South Korea, *C. robusta* is legally recognized and managed as a ''Marine ecosystem disturbing species'' (a legal term for marine harmful organisms; (*Kim et al., 2019b*)).

The life cycle of *C. robusta* has a substantial impact on the dynamics of the marine ecosystem through both trophic interactions and spatial competition. This unique life cycle affects ecosystem function in two main ways: (1) the planktonic larvae serve as a major food source for various invertebrates (*Bingham & Walters, 1989*; *Boltovskoy & Correa, 2015*; *Rivera-Figueroa et al., 2021*). (2) The widespread settlement and growth of adults lead to competition for space with other fouling organisms (*Grosberg, 1981*; *Bullard, Whitlatch & Osman, 2004*). Moreover, the reproductive and growth patterns of *C. robusta* can have a detrimental impact on local ecosystems through competition with native species. This highlights the necessity of comprehending its reproductive and growth dynamics for the implementation of effective management and conservation strategies (*Robinson et al., 2017*; *Part et al., 2018*).
There are at least two generations of *Ciona robusta* and *Ciona intestinalis* (type-B) per year in regions such as Naples and Brittany (*Caputi et al., 2015*; *Bouchemousse, Lévêque & Viard, 2017*). However, in warmer climates, reproduction is more intense, resulting in shorter generation periods for populations. For example, in tropical and subtropical regions, more spawning occurs in a year than that in cooler regions such as the sub-Antarctic (*Wilson, Murphy & Wyeth, 2022*), and overall growth is relatively faster (*Malfant et al., 2017*). On the other hand, a study observing oocyte size and gonad development in the cooler climate of Puerto Madryn port, Argentina (from 9 °C in winter to 19 °C in summer), found that reproduction occurs throughout the year, but abundance peaks only once in the fall (*Giachetti et al., 2022*; *Giachetti, Tatián & Schwindt, 2022*). The maximum growth length in Kyoto, Japan (a temperate region), has been recorded to be up to 130 mm (*Tarallo et al., 2016*). Growth also varies not only with temperature but also with feeding. In a laboratory study at a temperature of 15 °C, growth rates varied depending on the type of feed, with a maximum average growth of 11.59 mm in 32 days when fed appropriately (*Zupo et al., 2020*). However, these individual growth studies provide limited insight into population-level dynamics and temporal patterns in natural environments.

The *in situ* lifespan of *C. robusta* is reported to range from 2 months to 2 years (*Millar, 1952*; *Dybern, 1965*). Cohort-level analysis provides critical insights into population structure, reproductive timing, recruitment success, and although detailed information is limited, multiple generation overlap patterns that cannot be captured through bulk population assessments (*Rius, Pineda & Turon, 2009*; *Wagstaff, 2017*). For invasive ascidians, distinguishing individual cohorts are likely to coexist simultaneously depending on temperature enables precise quantification of growth rates under varying environmental conditions (*Yamaguchi, 1970*; *Yamaguchi, 1975*), identification of optimal reproductive windows, and assessment of population turnover rates (*Rosner & Rinkevich, 2024*). Previous studies focusing on seasonal abundance patterns without cohort resolution have limited our understanding of fine-scale population dynamics and their environmental drivers (*Mastrototaro, D'Onghia & Tursi, 2008*). This gap in knowledge hampers the development of targeted management strategies, as effective control measures require understanding of when and how rapidly new cohorts establish and mature (*Xu et al., 2013*).

Currently, understanding the cohort and growth patterns of these invasive ascidians is crucial for assessing their invasion success (*Wong, McClary & Sewell, 2011*; *Lynch et al., 2016*). Therefore, the first aim of this study was to closely examine the cohort patterns and growth rates of *C. robusta*, a representative marine invasive species. By correlating these patterns with environmental variables, notably water temperature and salinity, this research aims to unveil the intricate relationships between the lifespan of *C. robusta* and its surrounding ecosystem. Water temperature and salinity have been studied relatively more than other environmental factors (*e.g.*, substrate materials, light conditions and pH; *Li et al., 2019*; *Olivo et al., 2021*; *Jones, Holt & Chan, 2022*) for *C. robusta* and are known to have a decisive effect on *C. robusta* growth (*Wilson, Murphy & Wyeth, 2022*).

However, most previous studies on these factors were conducted under controlled laboratory conditions. These conditions may not fully represent the complex environmental interactions that occur in natural marine ecosystems, limiting our understanding of their

effects on *C. robusta* growth under *in situ* conditions. Thus, the second aim of this study was to determine whether temperature and salinity were the main factors affecting growth rates, even *in situ* environment with multiple variables. The findings of this investigation promise to not only increase the ecological understanding of *C. robusta* but also inform and refine strategies for their management and control. Through a detailed analysis of the cohort dynamics of *C. robusta* under different environmental conditions (especially water temperature), this study seeks to provide valuable insights into the broader challenge of conserving marine biodiversity in situations such as climate change, where the threat of invasive ascidians is increasing.

## MATERIALS & METHODS

In this study, ten surveys were conducted at Mokpo Yacht Marina (34°47′2.70″N, 126°23′21.05″E) at 2-week intervals from June 2022 to October 2022. The study site, Mokpo Yacht Marina, is located in Mokpo City, Jeollanam-do, South Korea (hereafter referred to as Mokpo; Fig. 1). The geographic images used in Fig. 1 were referenced from Google Earth 7.3. *Ciona robusta* was collected from the marina at depths of about one m. Sampling continued until October 29, 2022. *Ciona robusta* individuals were not collected in the ninth and tenth surveys, which were conducted on October 15 and October 29, 2022. The objective was to sample at least 50 individuals on each survey. However, due to a lack of live individuals, only 30 were sampled on the last survey (October 1, 2022), where *C. robusta* was last observed. Only undamaged individuals were used to measure body length during the sampling process. We also anesthetized the relaxed body by placing it in a plastic tray (300 × 240 × 70 mm) filled with local seawater and menthol crystals to prevent contraction. The total body length was measured using digital-type Vernier calipers (Digimatic Caliper; Mitutoyo Corporation, Kanagawa, Japan) and imaged using a camera (Tough TG-5; Olympus Corporation, Tokyo, Japan). The measured individuals were placed in 50-mL conical tubes and stored in a portable freezer ($-20$ °C) for transportation to the laboratory. In the laboratory, each specimen was weighed wet and then vacuum freeze-dried (FDT-8650, Operon, South Korea) for 72 h to measure its dry weight. Wet weight and dry weight were measured (accurate to 0.01 g) using an electric weighing scale (ML4002/01, Mettler Toledo, Switzerland). Water temperature and salinity data were provided by the Korea Hydrographic and Oceanographic Agency of Ocean Data in the Grid Service (https://www.khoa.go.kr, Fig. S1). To reduce the error in the body length data measured in the field, the body length data were calibrated by remeasuring the length in the image using ImageJ (*Schneider, Rasband & Eliceiri, 2012*; National Institutes of Health, Bethesda, MD, USA). In this study, body length data served as the primary metric for analysis. To validate the accuracy of body length data, dry and wet weights were also measured and subjected to correlation analysis (Fig. S2). This step ensured that all analyses were grounded on precise and reliable body length measurements, forming the basis of the study's findings.

The body length of the shortest individual collected was eight mm, and therefore the body length frequency distribution was consisted with seven mm intervals. Each cohort was

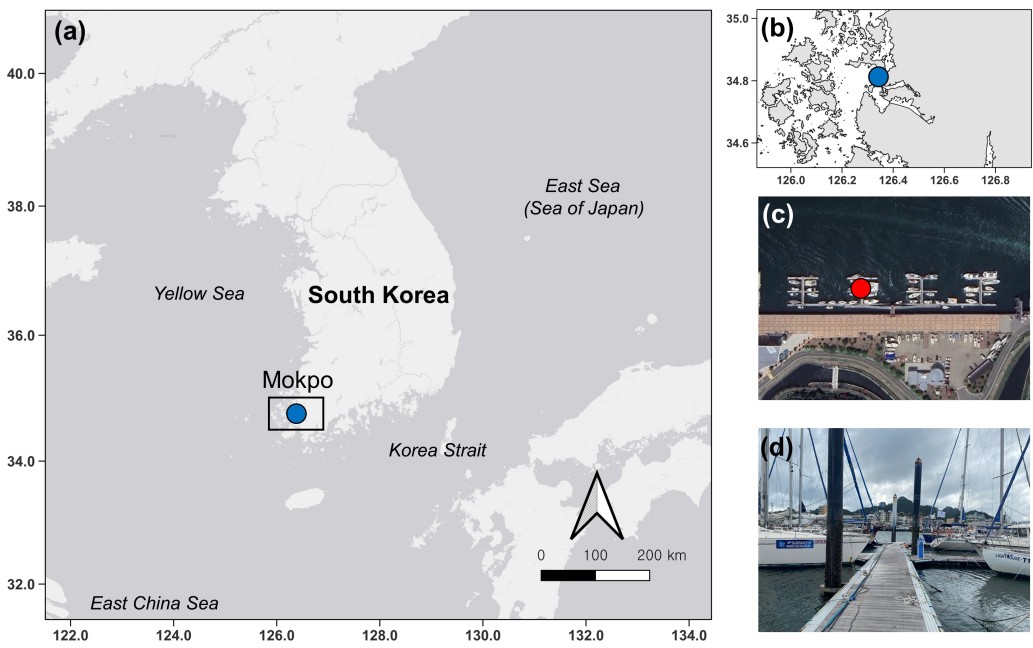

**Figure 1** **Map and photos of the survey site.** Map showing the location of the survey site (blue dot) in South Korea (A). Zoomed-in study site (B). Satellite imagery indicating points (red dot) where artificial substrates were installed (C) and foreground photo (B).

determined using the Bhattacharya method (*Bhattacharya, 1967*). FiSAT II software v 1.2.2 (*Gayanilo, Sparre & Pauly, 2005*; FAO, Rome, Italy) was used to separate the components of the normal distribution for each survey. The normal distribution was determined iteratively until it could no longer be distinguished using the separation index ((SI) = ΔLk/Δδk, where ΔLk is the difference between the two successive means of the component curves and Δδk is the difference between their estimated standard deviations). For SI values above 2, cohort separation was found to be statistically reliable. The growth rate of each cohort was calculated using the formula $r = (\ln N_{ti+1} - \ln N_{ti})/(t_{i+1} - t_i)$ (*Odum, 1971*), where '$r$' represents the growth rate of the population; 'N' represents the average body length of the cohort—*Choe & Deibel (2011)* used abundance but we used body length—'t' represents time, which in this study refers to the order of the survey; 'i' represents a specific point in time, which in this study refers to the number of surveys. We performed $t$-tests to compare mean body lengths between cohorts and multiple linear regression to determine whether the growth rate of cohorts was affected by environmental factors (water temperature and salinity). Growth rate was used as the dependent variable and water temperature and salinity as the independent variables, and the independent variables were tested for normality and equality of variance using the Shapiro–Wilk and Levene test, respectively. The $t$-tests and multiple linear regression were performed in R (*R Core Team, 2013*; R Foundation for Statistical Computing, Vienna, Austria). The $t$-test was performed by generating normally distributed data using the 'rnorm' function. We also used the 'lm' and 'summary' functions to perform multiple linear regression and model summary statistics,

**Table 1 Characteristics of *Ciona robusta* cohorts.** Population measurements and characteristics of cohorts (C1–C5) across different survey dates in 2022.

| Date | Survey number | Cohort | Mean body length (mm) | SD | Population | Separation index | $n$ |
|------|------|------|------|------|------|------|------|
| | | C3 | 44.25 | 12.21 | 44 | | |
| Jun. 26. 2022 | 1 | C2 | 72.22 | 4.52 | 10 | 3.34 | 62 |
| | | C1 | 99.06 | 14.91 | 8 | 2.76 | |
| | | C4 | 16.50 | 9.53 | 25 | | |
| Jul. 9. 2022 | 2 | C3 | 45.28 | 9.25 | 24 | 3.06 | 63 |
| | | C2 | 72.75 | 5.89 | 6 | 3.63 | |
| | | C1 | 95.62 | 8.34 | 7 | 3.21 | |
| | | C4 | 30.80 | 13.24 | 31 | | |
| Jul. 23. 2022 | 3 | C3 | 69.04 | 6.94 | 24 | 3.79 | 79 |
| | | C2 | 102.05 | 12.88 | 24 | 3.33 | |
| | | C4 | 41.52 | 14.91 | 38 | | |
| Aug. 5. 2022 | 4 | C3 | 90.07 | 13.06 | 12 | 3.47 | 58 |
| | | C2 | 135.43 | 15 | 8 | 3.23 | |
| | | C5 | 28.35 | 6.09 | 27 | | |
| Aug. 20. 2022 | 5 | C4 | 67.60 | 9.55 | 19 | 5.02 | 53 |
| | | C3 | 123.50 | 8.41 | 7 | 6.22 | |
| Sep. 3. 2022 | 6 | C5 | 37.39 | 10.32 | 25 | | 60 |
| | | C4 | 89.22 | 17.27 | 35 | 3.76 | |
| Sep. 17. 2022 | 7 | C5 | 50.34 | 16.40 | 64 | | 64 |
| Oct. 1. 2022 | 8 | C5 | 51.03 | 6.86 | 30 | | 30 |

respectively. Damaged individuals were identified based on morphological characteristics and excluded from sampling to prevent potential bias in measurement values (*Tamburini et al., 2022*).

# RESULTS

A total number of 469 *C. robusta* individuals were collected in this study, with a mean of $58.62 \pm 13.77$ (mean ± standard deviation (SD)) individuals collected in each field survey (Table 1). The shortest and longest individuals (collected on July 9 and August 5) were 8 and 150 mm, respectively. The *C. robusta* body length values measured for cohort analysis and growth rate calculations showed a significant positive correlation with wet ($r^2 = 0.81$) and dry weight ($r^2 = 0.78$). Wet and dry weights were also significantly positively correlated ($r^2 = 0.91$; Fig. S1). The maximum observed water temperature during the study period was 26.69 °C (August 26), and the minimum was 22.01 °C (June 27). Salinity was highest at 30.63 practical salinity unit (PSU) (July 13) and lowest at 19.70 PSU (September 9). The ranges for salinity and temperature were 10.92 PSU and 4.68 °C, respectively, indicating a larger range for salinity than for temperature (Fig. S2).

Five cohorts were identified over the entire study period using the Bhattacharya method (C1–C5). All cohorts had SI > 2 at each survey time and were significantly separated

($p < 0.05$; Table 1 and Table S1). C3 was the most observed cohort, with five surveys (June 26 to August 20), and C1 was the least observed cohort (June 26 to July 9), with two surveys. The most cohorts were observed on July 9 (C1–C4), with four, and the fewest were observed on September 17 and October 1, with one cohort each (C5). Three or more cohorts were consistently observed from the first (June 26) to the fifth survey (August 20), with the last cohort (C5) being the first observed in the fifth survey (Fig. 2). The longest cohort was C2 (measured on August 5), with a mean body length of 135.43 ± 15.00 mm, and the shortest cohort was C4 (measured on July 9), with a mean body length of 16.50 ± 9.53 mm. The cohort with the largest range between the maximum and minimum mean body length was C3, which grew from 44.25 ± 12.21 mm (June 26) to 123.50 ± 8.41 mm (August 20), a growth of approximately 79.25 mm. Excluding C1, which had a decrease in mean body length, the cohort with the smallest difference in mean body length was C5, which grew from 28.35 ± 6.09 mm (August 20) to 51.03 ± 6.86 mm (October 1st), a growth of approximately 22.68 mm (Fig. 3 and Table 1).

The highest growth rate in this study was 0.6241 for C4, which grew 14.3 mm in mean body length from July 9 (16.50 mm) to July 23 (30.80 mm). In contrast, C1, which lost 3.44 mm in mean body length from June 26 (99.06 mm) to July 9 (95.62 mm), had the lowest growth rate of −0.0353 (Table 1 and Table S2). To confirm which environmental variables (water temperature and salinity) had a significant effect on growth rate, we first checked the assumptions for multiple linear regression. Normality and homogeneity of variances were verified for the residuals of a preliminary model ($p > 0.05$ for both tests; Table S4). After confirming these assumptions were met, we proceeded with the multiple linear regression analysis. Multiple linear regression analysis revealed that temperature significantly affected growth rate ($p < 0.001$, coefficient = 0.079), while salinity showed no significant effect ($p = 0.114$, coefficient = −0.024). The model explained 43.8% of variance in growth rates (adjusted $r^2 = 0.438$, F(2,17) = 8.413, $p = 0.002$) (Table S3). The model summary statistics for the multiple linear regression had an $r^2$ value of 0.438, an $F$-value (2, 17) of 8.413, and a $p$-value of 0.002. Temperature and growth rate were positively correlated, with higher temperatures leading to higher growth rates, and the coefficient of determination was relatively high ($r^2 = 0.51$). On the other hand, salinity and growth rate were negatively correlated, and the coefficient of determination was noticeably lower ($r^2 = 0.18$; Fig. 4).

## DISCUSSION

Mokpo, South Korea, is a region where *C. robusta* has been present continuously from spring to fall (*Park et al., 2018*; *Bae et al., 2022*; *Lee et al., 2022*). Because of the temperate climate in the study area, several populations hatch at similar times within a year. Therefore, we conducted this study to examine the cohort patterns and growth rates of *C. robusta* and determine whether temperature and salinity are the main factors affecting growth under an *in situ* environment. A total of five cohorts were observed during the study period, and the SI values of all cohorts were >2, indicating that the cohorts were well distinguished. This follows the criteria of existing protocols and other studies, which consider values below

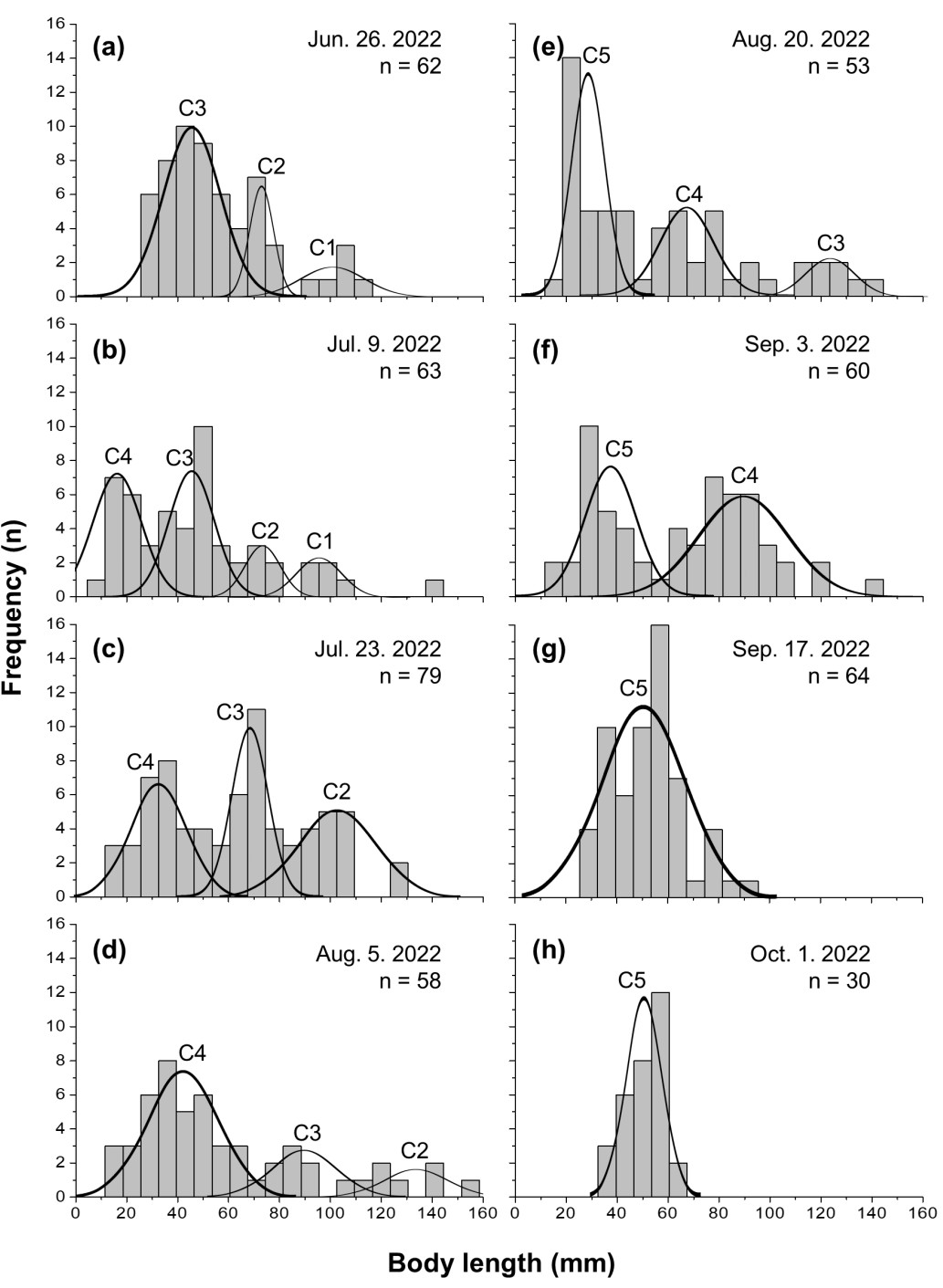

**Figure 2 Frequency distribution of body length in Ciona robusta.** Frequency distribution of body length in Ciona robusta samples collected between June 26, 2022, and October 1, 2022, numbered by 2-week interval (A–H). Individual cohorts were defined as the normally distributed components of the sample distribution.

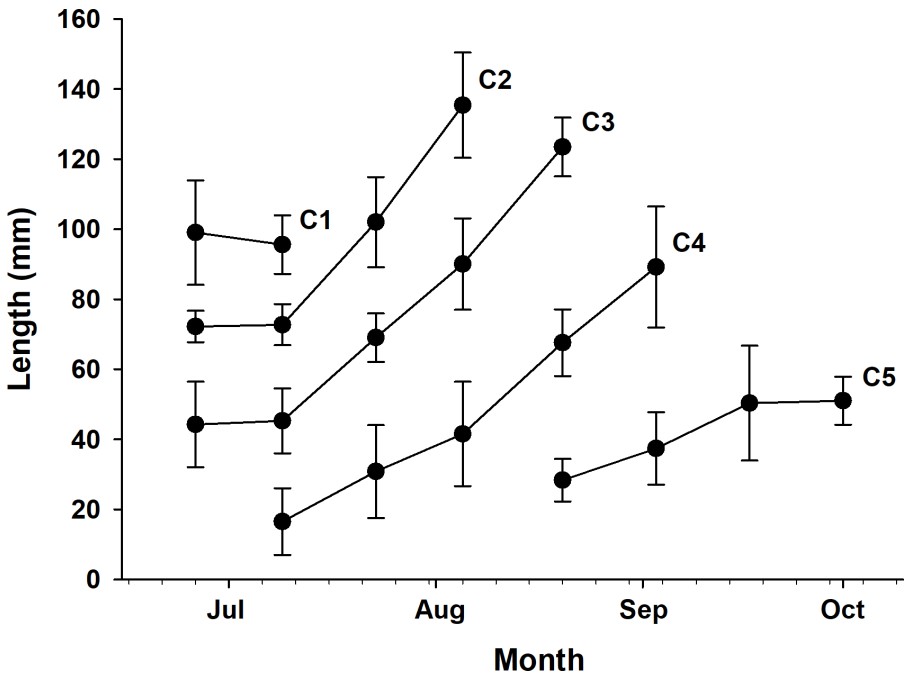

**Figure 3 Time series of mean body length.** Time series of mean body length (with standard deviation) for each cohort (1–5) observed over the study period.

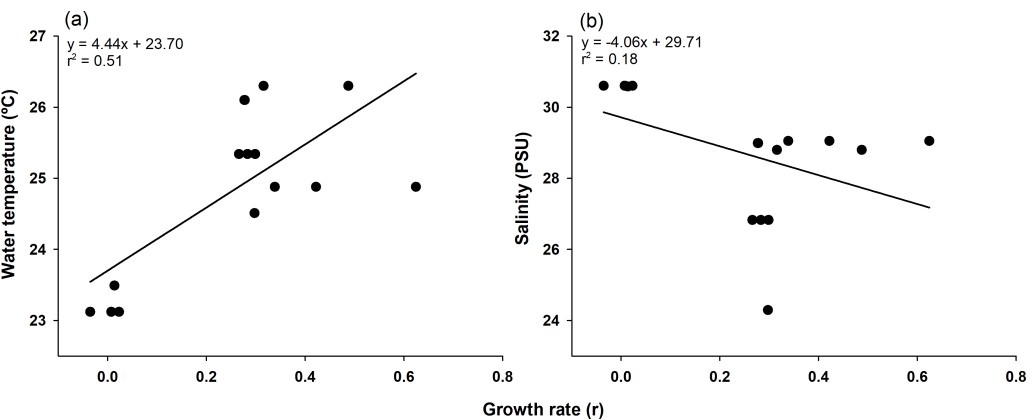

**Figure 4 Pearson correlation of growth rate with environmental factors.** Pearson correlation of growth rate with water temperature (A) and salinity (B) in *Ciona robusta*, where the black line represents the regression line.

2 to be unreliable. This follows existing protocols, which consider values below 2 to be unreliable (*Gayanilo, Sparre & Pauly, 2005*), and other studies that use values exceeding 2 as the criterion (*Arculeo et al., 2011*; *Lolas & Vafidis, 2021*).

At Mokpo, the number of cohorts separated within approximately the same duration of study periods was higher than in other areas. Many studies have documented the seasons when ascidians of the genus *Ciona*, dominate communities, but few have distinguished

individual cohorts. While *C. robusta* and *C. intestinalis* are now recognized as genetically and ecologically distinct species (*Mastrototaro, D'Onghia & Tursi, 2008*; *Astudillo, Leung & Bonebrake, 2016*; *Bouchemousse, Lévêque & Viard, 2017*), we reference both due to limited cohort-specific literature, acknowledging that direct species comparisons require caution. In Nova Scotia (Canada), up to two major recruitment events occur between June and August (*Carver, Chisholm & Mallet, 2003*). In Naples, Italy, three genetically distinct clusters (cohort) in the population were observed following monthly sampling for 13 months (*Caputi et al., 2019*). In comparison, Naples and Nova Scotia showed fewer distinct cohorts than observed in Mokpo.

The cohort lifespan of genus *Ciona*, such as *Ciona robusta* (*Nakazawa, Shirae-Kurabayashi & Sawada, 2019*; *Beyer et al., 2023*) and *Ciona intestinalis* (*Millar, 1952*; *Dybern, 1965*; *Beyer et al., 2023*), varies from 2 months to 2 years. The observed cohort lifespan in this study (4–10 weeks) falls within the shorter range of previously reported values, consistent with field studies showing lifespans of approximately 3 months (*Nakazawa, Shirae-Kurabayashi & Sawada, 2019*; *Beyer et al., 2023*). The growth, reproduction, and mortality of *C. robusta* are affected by various environmental factors, including natural (temperature, salinity, and microalgae) and anthropogenic (bisphenol A, mercury, copper, and cadmium) factors (*Bellas, Vázquez & Beiras, 2001*; *Bellas, Beiras & Vázquez, 2004*; *Mansueto, Cangialosi & Faqi, 2011*; *Wilson, Murphy & Wyeth, 2022*).

For body length, the maximum body length of the *C. robusta* we sampled was 150 mm, with C2 having the longest mean body length of the cohort at 135.43 ± 15.00 mm. These results are consistent with findings from nearby temperate regions, such as the maximum body length of 130 mm recorded in Kyoto, Japan (*Tarallo et al., 2016*).

During the study period, five cohorts were observed, suggesting that two or more cohorts existed at different points in time simultaneously. The observation of multiple overlapping cohorts can be indirectly supported by the extended reproductive capacity of *C. robusta*, as demonstrated by the annual gonadal maturation reported in previous studies (*Giachetti, Tatián & Schwindt, 2022*). Although direct comparisons with other studies were not performed, the observed differences in cohort number and lifespan may be attributed to unmeasured environmental factors (*e.g.*, food availability and predation pressure) or differences in analytical methodologies. Specifically, live algae and non-live particles ensure the survival of *C. robusta* (*Zupo et al., 2020*); therefore, these environmental factors may have played a role or previous monitoring studies may not have used a normal distribution to separate cohorts. A wide range of information is available on post-settlement growth rates of juveniles, but linking the information to understand the factors that influence them remains a challenge (*Wilson, Murphy & Wyeth, 2022*). However, in this study, the growth rates of the separate cohorts enabled us to quantify the effects of temperature and salinity on *C. robusta* growth, with our statistical analysis demonstrating a significant positive relationship with temperature ($p < 0.001$) but no significant relationship with salinity ($p = 0.114$). Therefore, we analyzed the correlation between growth rates and two environmental factors (temperature and salinity). Based on our statistical results, temperature had a stronger effect than salinity among the environmental variables examined in this study. The multiple linear regression results showed that the growth

rate of each *C. robusta* cohort was more relatively correlated with temperature than with salinity, consistent with our findings that *C. robusta* development is most closely associated with increases in temperature (*Yamaguchi, 1970*; *Yamaguchi, 1975*).

*C. robusta* is relatively intolerant of low salinity conditions and has a developmental salinity limit of 26 ‰ (*Madariaga et al., 2014*; *Kim et al., 2019b*). Outside of these low salinity conditions, *C. robusta* will grow rapidly up to (and beyond) 24.7 °C in the laboratory (*Kim et al., 2019b*). Notwithstanding, the optimal temperature for development *in situ* was determined to be 14.7–23.7 °C (*Caputi et al., 2019*). It is presumed that the differences between laboratory and field results are due to additional environmental factors such as food availability or acclimation effects (*Zupo et al., 2020*; *Mathiesen et al., 2025*). Therefore, the temperature (24.88 °C) and salinity (29.05 PSU) conditions for C4 on July 23, when the growth rate was highest (0.6241), were optimal for *C. robusta* to achieve rapid growth. Correlations with growth rate performed using the Scheirer–Ray–Hare test on *C. robusta* juveniles in the laboratory under four conditions, two each of temperature (12 and 17 °C) and salinity (25 and 35 PSU) in combination, demonstrated statistical significance for all sources ($p < 0.05$; *Malfant et al., 2017*). However, in the present study, only temperature significantly positively correlated with growth rate. These results are attributed to the salinity measured during the study not persisting below the low salinity limit of 26 PSU. Although conditions below 26 PSU existed in the field, they were relatively short-term, lasting approximately 2 or 7–12 days compared with the 28 days or more in a laboratory study (*Malfant et al., 2017*). Therefore, the impact of low salinity may have been minimal.

Overall, the current study identified five *C. robusta* cohorts, each with a significant SI, indicating clear distinctions. The number of cohorts identified in Mokpo (five) was higher than those reported in other regions, though the factors contributing to these differences remain unclear. The lifespan of the *C. robusta* cohort in Mokpo is shorter than previously recorded (up to two years; *Millar, 1952*; *Dybern, 1965*), estimated to be between 4 and 10 weeks, contrasting with previous studies showing longer lifespans. In addition, growth rates were more strongly correlated with temperature than salinity, highlighting the important role of temperature.

Particularly encouraging is the fact that the correlation between cohort growth rate and temperature was demonstrated *in situ* rather than in laboratory environments, providing insights into growth responses under natural environmental variability. Salinity had a less significant correlation with growth rate compared with that of temperature, but this is likely because low salinity did not persist long enough at the study site to significantly affect growth rate. However, because this study was limited in geographic and temporal scope and focused primarily on temperature and salinity, it may have overlooked other environmental factors that could affect the growth and spread of *C. robusta*, such as ocean acidification and nutrient levels. Further research should include long-term observations in more diverse geographic locations, incorporating a wider range of environmental variables, and exploring correlations.

## CONCLUSIONS

Through an *in situ* study, we observed multiple cohorts thriving simultaneously in a Mokpo, South Korea and concluded that temperature strongly influences the growth rate of *C. robusta*. By elucidating the cohort patterns and temperature-dependent growth rates of *C. robusta*, this study provides useful information for understanding *C. robusta* population dynamics and for management strategies. This understanding is particularly valuable in the context of climate change, as shifts in temperature could alter the invasion dynamics and ecological impacts of *C. robusta*. This study highlights the need for climate zone-specific monitoring *C. robusta*. The insights gained from this study contribute to our understanding of *C. robusta* population ecology and may inform management approaches to effectively respond to ecosystem disturbances induced by climate change, for this species under changing environmental conditions. In future research, a broader understanding of the *C. robusta* cohort would be gained if long-term surveys were conducted in multiple geographic locations (varying climates), including factors such as ocean acidification and nutrient levels that were not addressed in this study.

### Funding

This work was supported by a grant from 'National Marine Biodiversity Institute of Korea' (2025M00300). The funders had no role in study design, data collection and analysis, decision to publish, or preparation of the manuscript.

### Grant Disclosures

The following grant information was disclosed by the authors:
National Marine Biodiversity Institute of Korea: 2025M00300.

### Competing Interests

The authors declare there are no competing interests.

### Author Contributions

- Philjae Kim conceived and designed the experiments, performed the experiments, analyzed the data, authored or reviewed drafts of the article, and approved the final draft.
- Seongjun Bae conceived and designed the experiments, performed the experiments, analyzed the data, prepared figures and/or tables, and approved the final draft.

### Data Availability

   The raw measurements are available in the Supplementary File.

### Supplemental Information

Supplemental information for this article can be found online at http://dx.doi.org/10.7717/peerj.20034#supplemental-information.

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
