# Peer review of "In situ study of environmental factors (temperature and salinity) affecting cohort patterns and growth rates in Ciona robusta"

_PeerJ, doi:10.7717/peerj.20034_

## Round 0.1 · original submission · Major Revisions

I have big concern when one of the reviewer given a reject as their recommendation. The authors must able to satisfy the requirements and each comments from reviewers especially concerning to new contribution of knowledge and the concerns of misinterpretation/overinterpretation of data as compared to previous studies done.

·

Basic reporting

There are multiple instances of unclear and ambiguous phrasing throughout the manuscript, which affects the clarity and readability of the text. In addition, many of the citations are accompanied by interpretive statements that appear to overextend or misrepresent the original findings. In particular, some references are used to support claims (e.g., related to ecosystem-level impacts or generation number across geographic regions) that are either not directly addressed or only partially supported in the cited literature. This weakens the contextual foundation of the study and calls for a more careful and accurate representation of prior research.
The structure of the Introduction follows a general progression from the broader issue of marine invasive species, to general characteristics of ascidians, to the taxonomy and biology of Ciona robusta, and finally to the stated aims of the study. However, the logical connection between successive paragraphs is weak. It is unclear how each paragraph builds upon the previous one, and the transitions do not effectively guide the reader through the rationale leading to the research objectives.
The Methods and Results sections are clearly written and present the procedures and findings in a straightforward and factual manner. No major issues were noted in these sections.

Experimental design

While the manuscript may fall within the aims and scope of PeerJ, it suffers from a critical flaw as a scientific paper: the research question is poorly justified. The authors focus on the effects of salinity and temperature, yet in the Introduction, they themselves state that these variables have already been extensively studied in previous research. If these factors are already well understood, the authors must provide a compelling and logical rationale for re-examining them, especially in preference to less-studied environmental variables. Without such justification, the study lacks novelty and scientific urgency.
Furthermore, both the Introduction and Discussion reveal that the relationship between salinity, temperature, and growth rate in C. robusta has been addressed in numerous previous studies. The manuscript fails to clearly identify what specific gaps in the existing literature this study addresses or resolves. Unless the authors can articulate what this study achieves that prior work did not, the manuscript cannot be considered for publication.
Substantial revision of the Introduction is therefore required. And since the Discussion and Conclusions are logically dependent on the framing of the Introduction, they too must be significantly revised in accordance.

Validity of the findings

If the authors intend to emphasize the significance of conducting an in situ study on Ciona robusta, it is essential that they clearly distinguish their work from previous research. Notably, relevant in situ studies on Ciona intestinalis (e.g., Petersen et al. 1997) and C. robusta (e.g., Emmerson et al. 2023) are not cited or discussed in the manuscript. These omissions weaken the framing of the study’s novelty. Moreover, as the authors themselves acknowledge, numerous studies have already investigated the effects of temperature and salinity on growth in C. robusta. Therefore, to justify the current study’s contribution, the Introduction should explicitly explain what this study achieves that previous in situ or environmental studies have not, particularly in terms of methodological approach, spatiotemporal scale, or analytical depth.

Petersen et al. 1997: https://www.sciencedirect.com/science/article/abs/pii/S0022098197000646
Emmerson et al. 2023: https://www.journals.uchicago.edu/doi/full/10.1086/719476

The conclusions drawn in this manuscript significantly exceed the bounds of the study and are not scientifically defensible in their current form. Although the research is limited to Ciona robusta, the authors extend their findings to invasive species in general, making broad claims regarding predictive modeling, management practices, and climate change implications. These extrapolations are not supported by the data. The study lacks any comparative analysis with other taxa and provides no discussion of interspecific variation or similarities that would warrant such generalization. Unless the conclusions are comprehensively revised to align with the actual scope of the research—and any broader implications are supported by appropriate comparative evidence—the manuscript cannot be considered scientifically sound or suitable for publication.

Additional comments

The manuscript contains too many instances of misrepresenting or overinterpreting previous studies, including citing sources that either do not support the claims made or do not address the topics in question at all. This practice constitutes a serious breach of scientific rigor. Misleading or unsupported citation undermines the credibility of the manuscript and cannot be overlooked in a peer-reviewed publication. The authors must thoroughly review each reference to ensure that citations are accurate, appropriate, and scientifically justified. As it stands, this issue represents a critical flaw that must be addressed for the manuscript to be considered for publication.

Please refer to the attached PDF for specific comments on the points that I believe require revision in the manuscript.

·

Basic reporting

The manuscript provides a clear and detailed account of the study on Ciona robusta, with a well-defined title and abstract that succinctly capture the focus on environmental drivers (temperature/salinity) and cohort dynamics. The introduction effectively reviews relevant literature, including ecological impacts and taxonomic reclassification, though some redundancy exists (e.g., repeated mentions of lifespan ranges Lines 89/90). Methods are thoroughly described, including cohort separation techniques and statistical analyses, but lack discussion of potential sampling biases (e.g., exclusion of damaged individuals). Results are presented logically, with growth rates linked to temperature, though salinity’s ecological relevance could be further explored. The discussion should better reconcile the field-observed optimal growth temperature (24.88°C) with the narrower lab range (14.7–23.7°C), possibly addressing factors like food availability or acclimation effects.
Overall, reporting is robust but would benefit from minor refinements (e.g., grammatical corrections like "C. robusta were collected" instead of was collected Line 115) and explicit statements on methodological approaches in the abstract.
The figures and the table are relevant to the study, presenting data clearly and with high-quality visuals. All figures are well-labeled and described in the manuscript, aiding in the interpretation of results. However, there are some points are not clear: 1. the map (figure1) lacks a proper source citation (e.g., Google Earth, or other geographic data providers). Including this reference would enhance reproducibility and align with standard academic practices. Other than this minor omission, the figures effectively support the study’s findings and contribute to the overall clarity of the research. 2. The title of figure4 "Pearson correlation of growth rate with water temperature (a) and salinity (b), where the black line represents the regression line" is well-constructed and effectively illustrates the statistical relationships. However, the title omits the species name (Ciona robusta), which is essential for clarity and scientific precision. Including the species name (e.g., "Pearson correlation of growth rate with water temperature (a) and salinity (b) in Ciona robusta...") would ensure consistency with the rest of the manuscript and avoid ambiguity.

Experimental design

The study design is rigorous, employing in situ field surveys and the Bhattacharya method to distinguish cohorts, validated by length-weight correlations. However, key limitations include the absence of replication details and potential sampling bias toward undamaged individuals, which may underrepresent smaller cohorts. The choice of 2-week survey intervals is not explicitly justified—whether based on pilot data or prior studies—raising questions about temporal resolution. While statistical analyses (e.g., multiple linear regression) are appropriate, the non-significance of salinity (p=0.114) warrants deeper ecological discussion. Strengths include the use of supplementary materials for reproducibility, but the design could be strengthened by addressing sampling representativeness and clarifying interval selection criteria.

Validity of the findings

The findings are compelling, particularly the identification of five distinct cohorts and the strong correlation between growth and temperature, supported by statistical significance (p<0.001). The shorter lifespan observed (4–10 weeks) aligns with field conditions but contrasts with lab studies, a discrepancy well-justified by anthropogenic stressors. However, the reported "optimal temperature" (24.88°C) slightly conflicts with lab ranges (14.7–23.7°C), suggesting a need to reconcile these differences—possibly via factors like food availability. Salinity’s non-significance is appropriately downplayed, though its transient effects merit discussion. The conclusions are valid and impactful, but broader implications (e.g., climate change effects on invasion rates) could be emphasized. Future research directions (e.g., ocean acidification) are well-identified, though management recommendations (e.g., temperature-focused monitoring) would enhance practical relevance.

Additional comments

The manuscript requires minor revisions to align with standard academic formatting conventions:
1. Species nomenclature: The scientific name Ciona robusta (line 231) should be italicized to comply with taxonomic formatting standards.
2. Citation formatting: In-text citations should consistently use a comma to separate the author(s) and year (e.g., "Author, Year") in accordance with journal style guidelines. Please ensure all citations follow this format throughout the document.

---

## Round 0.2 · Minor Revisions

Please refer to the annotated manuscript and I am happy to accept this paper after some minor revision.

·

Basic reporting

The manuscript contains some inaccuracies, such as the use of the term "high biodiversity" and the conceptual treatment of Ciona robusta as a species. However, the structure of each paragraph has been reorganized and the arguments within the manuscript have become more consistent, resulting in a clearer articulation of the authors' main claims.

Experimental design

The manuscript can be considered a strong one, as it clearly demonstrates the significance of investigating cohort dynamics in situ and examining their relationship with seawater temperature and salinity. The importance of the research question has been clarified, and the distinction between this study and previous research has been more clearly articulated.

Validity of the findings

The Conclusion is based on the content presented in the Introduction, Methods, Results, and Discussion, and clearly states the claims in relation to the objectives of the study.

Additional comments

The manuscript does not appear to have any major issues. However, some revisions are needed to address potentially exaggerated expressions and to clarify the definitions of certain terms and concepts. Please refer to the attached file, in which I have provided specific comments.

---

## Round 0.3 · Minor Revisions

We are happy to accept after slight comments from reviewer are addressed

---

## Round 0.4 · accepted · Accept

I am happy all of the concerns were addressed very well and authors did a good job justifying each changes and also correction.